# *"You just forget about preeclampsia and move on"* –awareness of chronic disease risks and follow-up preferences after preeclampsia in Ireland: a national qualitative study

Peter M. Barrett[1,2,3]*, Aisling Jennings[4], Heike Roth[5], Molly Byrne[6], Emma Wallace[4], Amanda Henry[5,7,8], Fergus P. McCarthy[1,9], Karolina Kublickiene[10], Ali S. Khashan[1,2]

**1** INFANT Research Centre, University College Cork, Cork, Ireland, **2** School of Public Health, University College Cork, Cork, Ireland, **3** Department of Public Health HSE South West, St. Finbarr's Hospital, Cork, Ireland, **4** Department of General Practice, University College Cork, Cork, Ireland, **5** Faculty of Health, University of Technology Sydney, Sydney, Australia, **6** School of Psychology, University of Galway, Galway, Ireland, **7** Discipline of Women's Health, School of Clinical Medicine, Faculty of Medicine and Health, University of New South Wales, Sydney, Australia, **8** The George Institute for Global Health, University of New South Wales, Australia, **9** Department of Obstetrics & Gynaecology, Cork University Maternity Hospital, Cork, Ireland, **10** Department of Clinical Science, Intervention and Technology, Karolinska Institutet, Stockholm, Sweden

* peter.barrett@ucc.ie

## Abstract

### Background

Preeclampsia is associated with increased long-term risks of cardiovascular disease, kidney disease, and stroke. International guidelines recommend structured follow-up care to prevent chronic disease, but limited research has explored women's knowledge of these risks, and their preferences regarding long-term follow-up care. This may impede how obstetric information is used for chronic disease prevention in practice.

### Methods

This qualitative study used purposive and snowball sampling to recruit women in Ireland diagnosed with preeclampsia at least one year prior. Semi-structured interviews were conducted online, exploring awareness of chronic disease risks and provision of follow-up care. Thematic analysis was performed using an inductive approach.

### Results

Twelve women aged 28–64 years were interviewed, at median six years since preeclampsia diagnosis. Participants' antenatal and postnatal care experiences varied widely, but most described follow-up care after preeclampsia as being inconsistent or absent. Three key themes were generated: (1) *Preeclampsia in the 'rear-view*

**Data availability statement:** Data cannot be shared publicly because the research participants only consented to their data being processed for the explicit purposes outlined in the study protocol which was approved by the Clinical Research Ethics Committee of the Cork Teaching Hospitals (ECM 4 (j) 10/09/2024). Any requests to access the data would need to be considered by the the Clinical Research Ethics Committee of the Cork Teaching Hospitals (crec@ucc.ie) for those who meet criteria for access to confidential data.

**Funding:** This research was funded by the Health Research Board Clinician Scientist Postdoctoral Fellowship Scheme in Ireland (www.hrb.ie; grant number CSF-2023-006). Dr. Barrett receives funding for protected academic research time through this scheme. The funders played no role in the study design, data collection and analysis, or preparation of the manuscript.

**Competing interests:** The authors have declared that no competing interests exist.

*mirror'*—women often viewed preeclampsia as an acute, resolved event, and had limited awareness of any long-term risks. However, they regarded chronic disease risk information as valuable and empowering; (2) *Changing priorities as 'life takes over'*— women often prioritised other family members' health needs above their own, particularly in the newborn phase. They favoured delaying discussions about chronic disease risks, ideally to 6–12 months after pregnancy, preferably provided through primary care; (3) Desire for proactive, *'blameless'* follow-up care—women favoured systematic, non-judgmental follow-up programmes underpinned by clear communication between obstetric and primary care services, continuity of healthcare providers, and free access. Some described residual anxiety relating to their preeclampsia experience, and emphasised the importance of sensitive, person-centred follow-up care.

## Conclusion

Women affected by preeclampsia in Ireland typically have limited awareness of its links with long-term chronic disease risks, and frequently experience a lack of structured follow-up care. They expressed strong support for receiving personalised information about opportunities for secondary prevention, and advocated for systematic, proactive follow-up. Participants emphasised that future models of care should be pragmatic, person-centred, and include default enrolment for all women.

---

## 1. Background

Preeclampsia has been linked to increased risk of chronic disease in later life, including cardiovascular disease, kidney disease, stroke and vascular dementia [1–4]. Women affected by preeclampsia are at increased risk of subclinical atherosclerosis, which persists in the years following pregnancy, and this suggests that early follow-up after pregnancy may confer long-term health benefits [5]. A known history of preeclampsia may provide new sex-specific opportunities for chronic disease prevention over the life-course [6], and international guidelines now suggest that this should influence therapeutic decision-making for cardiovascular disease prevention, including earlier initiation of statin therapy [7].

Clinical guidelines on preeclampsia typically emphasize that affected patients and their general practitioners (GPs) should be informed about links to future risk of chronic disease [6, 8]. However, survey-based research from Australia suggests that awareness of these links may be suboptimal among affected women [9]. This may be due to discrepancies between patient preferences and healthcare providers' perceptions regarding importance of discussing long-term health risks after pregnancy. Some affected women appear to value personalised information about their future risk of chronic disease [10]. However, healthcare professionals may underestimate the relevance of providing such information, particularly if there is uncertainty around how women should be followed up [11].

The optimal timing, format, and content of follow-up programmes for women affected by preeclampsia remain uncertain. At a population level, the relative risks of long-term cardiovascular disease and chronic kidney disease are approximately doubled in women affected by preeclampsia, compared with those who remain normotensive in pregnancy, but individualised risk may remain reasonably low [1, 2]. The feasibility and uptake of any post-pregnancy interventions depend largely on acceptability to patients, perception of long-term risk, and cultural factors [6]. In 2023, the publicly funded National Chronic Disease Prevention Programme in Ireland commenced systematic, annual monitoring after pregnancy, through primary care settings, for all women newly diagnosed with preeclampsia [12]. Although an estimated 5% of deliveries in Ireland are affected by preeclampsia every year [13], the exact number of women eligible for follow-up is unknown, partly due to the absence in Ireland of a common, embedded electronic health record [14]. Thus, the uptake of this expanded Prevention Programme, incorporating preeclampsia, is uncertain.

Internationally, preeclampsia remains highly prevalent, yet there has been limited research exploring affected women's knowledge of chronic disease risk, and their preferences regarding long-term follow-up care. This is likely to impede how obstetric information can be harnessed for chronic disease prevention in practice. The aims of this qualitative study are to explore awareness of long-term chronic disease risks among women affected by preeclampsia in Ireland, and to understand their perspective regarding the structure and availability of follow-up care after pregnancy.

## 2. Methods

### 2.1 Study design

A qualitative research design using a thematic analytical approach was used. Women aged 18 or older were invited to participate in qualitative interviews if they had been diagnosed with preeclampsia in Ireland on at least one occasion, at least one year prior to study participation (no upper time limit). To achieve maximum sampling variation, we employed a combination of purposive and snowball sampling to recruit women differing in age, geographic location (urban/rural), and time since preeclampsia diagnosis. The Republic of Ireland has a mixed public-private healthcare system, and we also sought to include a mix of participants who had experienced public and private obstetric care respectively. Specifically, the experience of obstetric care (including early follow-up care) may differ by health insurance status [15]; private antenatal care is led by a named Consultant Obstetrician, who provides high levels of continuity of care to each individual patient, whereas routine public antenatal care is typically midwife-led without individual access to a named Consultant. The latter sometimes involves continuity of care with the same midwife, but this is not routinely available for all. Similarly, post-partum care may vary from one individual to the next, since some components are standardised for all (e.g., home visits by public health nurses during first week post-partum), whereas others depend on patient initiative (e.g., making appointments for GP review at 6 weeks) or health insurance status (e.g., post-partum review by Consultant Obstetrician at 4–6 weeks).

Participants were invited to one-to-one, semi-structured interviews, scheduled at a time of their convenience. Semi-structured interviews were chosen to explore participants' awareness, thoughts and beliefs, while facilitating discussion of personal and potentially sensitive issues [16]. All interviews were facilitated by the lead researcher (PMB) and were offered online or in person.

### 2.2 Participant recruitment

Participant recruitment took place between 18/09/24 and 18/11/24. Recruitment posters were promoted through institutional social media accounts (Instagram, Linkedin, X) by the INFANT Research Centre, University College Cork, and by individual members of the study team through their own public social media accounts. A patient and public involvement (PPI) panel, composed of five women with lived experience of preeclampsia in Ireland, was established to advise on the study, and these contributors assisted in study promotion through their own personal networks. The PPI panel members

had responded to a public-facing advertisement through the PPI Ignite website in Ireland (ppinetwork.i.e.,), and comprised of women who represented a mix of ages, parity, geographical location in Ireland, and public/private health insurance status.

Potential study participants were invited to express interest via a QR code and a link to Microsoft Forms on the recruitment posters, or by contacting the lead researcher directly via email. They were considered eligible if they met the pre-defined inclusion criteria: diagnosis with preeclampsia in the Republic of Ireland at least once, more than one year prior, and currently residing in the Republic of Ireland. Any eligible women received a participant information leaflet, and follow-up contact was made to address questions and confirm whether they wished to proceed to interview. None of the participants were personally known to the researchers. All participants opted for online interviews, and the principles of information power were used to determine when a sufficient number of interviews had been conducted, after which no further participants were recruited [17].

## 2.3 Data collection

At the beginning of each interview, the lead researcher introduced himself and outlined the reasons for doing the research. A topic guide was used to elicit free-flowing discussion related to (i) participants' experience of follow-up care after pre-eclampsia, if any (ii) existing awareness of chronic disease risks (iii) preferences regarding future follow-up care, if any (Supporting Information S1 File). These topics had been agreed between the lead researcher and PPI contributors in advance of the first interview. Where possible, prompts were used to encourage elaboration on key points. Online interviews were auto-recorded, and automated transcription was employed based on in-built software in Microsoft Teams. All transcripts were verified and corrected by the lead researcher immediately after interviews had been completed. Transcripts were pseudonymised in order of study recruitment. Hand-written notes were also recorded during and immediately after each interview to capture more nuanced information and non-verbal cues, and these notes were included in the analysis.

## 2.4 Analysis

Transcripts were analysed using Braun & Clarke's Framework for Thematic Analysis to identify semantic and latent themes [18]. An inductive approach was used for analysis. Transcripts were first read repeatedly to gain overall familiarity with the data, and early impressions were noted. Open coding was used to develop and modify initial codes. Related codes were then grouped together to form categories, which were continuously reviewed to identify and generate themes.

The lead researcher coded all transcripts. Two other researchers (AJ, HR) independently coded two transcripts each. These were reviewed with the purpose of facilitating a reflexive discussion, and to allow for interpretive alignment and dialogue between the researchers.

Sharing qualitative research findings with participants can enhance credibility and trustworthiness of the findings [19, 20]. We undertook collaborative member reflection with three selected study participants after a preliminary analysis had been completed. In these one-to-one, online meetings with the lead researcher, the initial results were presented, openly discussed, and informant feedback and reflections were sought on the findings. Participants were invited to comment on whether the preliminary findings reflected their own feelings and experiences of follow-up care. These meetings led to some additional insights, including further refinement of our analysis to capture more nuanced, personal experiences and latent themes. The notes from these three meetings were subsequently included in the analysis, and influenced the final themes.

## 2.5 Reflexivity

The lead researcher (PMB) is a male medical doctor and postdoctoral researcher working in a joint clinical academic role within the Irish public health service, and in a university-based research organisation. His main area of research is maternal health, but he may have introduced medical preconceptions to the research process, or his role may have introduced

particular interview dynamics arising from gender-based or educational differences with participants. This is particularly relevant given the sensitive nature of this research, and since he was the sole interviewer. Nonetheless, comparison and discussion of coding between researchers (PMB, AJ, HR) allowed for triangulation of findings, and was used as a means of fostering reflexivity, since the other researchers were, respectively, a practising academic GP and qualitative researcher based in Ireland (AJ), and a practising academic midwife based in Australia (HR), both of whom are female.

## 2.6 Ethical Considerations

Participants completed an electronic informed consent form prior to interview. We anticipated that some participants may be unaware of links between preeclampsia and future risk of chronic disease, and that they may learn this information for the first time during interviews. Where women had been unaware of links between preeclampsia and chronic disease, they were provided with this information during the interview, reassured of the modifiable nature of any individual risk, and signposted to further information and supports if needed. A distress protocol was prepared to address potential participant upset but did not need to be activated during any interview. Ethical approval for the study was received from the Clinical Research Ethics Committee of the Cork Teaching Hospitals (ECM 4 (j) 10/09/2024).

## 3. Results

Twelve women were interviewed with median age of 39 years, at median 6 years after their preeclampsia diagnosis (Table 1). The interviews averaged 31 minutes duration (range 24–40 minutes). Participants lived in nine different Irish counties; seven participants lived in urban or suburban areas, and five lived in rural areas. Three participants had already been diagnosed with long-term health conditions since their preeclampsia diagnosis. Thirteen other women who had expressed interest in the study either did not meet eligibility criteria or were unable to attend interviews.
We did not ask participants directly about parity, nor occupational status. However, it became clear during interviews that participants self-identified as a mix of primiparous and multiparous women, and that three of them were healthcare professionals. It also became clear that participants' antenatal experiences had varied considerably. Some women had been very satisfied, and *"couldn't fault"* [P12] their antenatal care. Others reported disappointment or dissatisfaction with their antenatal care, and it is possible that these experiences impacted their views of post-pregnancy follow-up. Selected examples of responses to core questions from the topic guide are summarised in Supporting Information S2 File. Although

**Table 1. Characteristics of interview participants.**

|  | Age | Years since diagnosis | Obstetric care | Residence | Long-term condition(s)* |
|---|---|---|---|---|---|
| **P1** | 39 | 9 | Public | Urban/ Suburban | |
| **P2** | 38 | 2 | Private | Rural | |
| **P3** | 64 | 42 | Private | Urban/ Suburban | Chronic hypertension, Cardiovascular disease |
| **P4** | 28 | 5 | Public | Rural | |
| **P5** | 34 | 1 | Public | Rural | Chronic hypertension |
| **P6** | 38 | 2 | Semi-private | Urban/ Suburban | |
| **P7** | 38 | 7 | Public | Urban/ Suburban | |
| **P8** | 39 | 9 | Public | Urban/ Suburban | |
| **P9** | 40 | 1 | Private | Urban/ Suburban | |
| **P10** | 48 | 15 | Public | Rural | |
| **P11** | 45 | 7 | Public | Urban/ Suburban | |
| **P12** | 31 | 1 | Public | Rural | Chronic kidney disease |

*diagnosed in the intervening years since their preeclampsia diagnosis

we did not ask participants directly about severity of preeclampsia, some disclosed that they, or their baby, had experienced adverse outcomes.

Three themes were generated from the data: (1) Preeclampsia in the *'rear-view mirror'*, (2) Changing priorities as *'life takes over'*, (3) Desire for proactive, *'blameless'* follow-up care.

### 3.1 Preeclampsia in the *'rear-view mirror'*

When participants were discharged from the maternity hospital, some recalled an immediate shift in the intensity of post-natal monitoring. Some participants who had experienced prolonged medical monitoring antenatally perceived that their postnatal discharge felt abrupt, or that they *"got dropped"* [P5]. They were surprised not to receive more extensive after-care, leaving them with an impression that preeclampsia was only an acute issue relevant during pregnancy, but which was now *"in the rear-view mirror"* [P7]. As their blood pressure normalised after discharge, some women described how they began to *"forget about preeclampsia and move on"* [P6] from the diagnosis, becoming focused instead on their new-born baby's needs. They suggested that the messaging received from the maternity service, while intended to be reassuring, probably consolidated these views.

*"Once the baby's out, and we know that Mum is somewhat stable, we're in the all clear… it's kind of like, okay, that's done with. Case closed, we're good, they've survived."* [P4]

Almost all participants described their long-term follow-up care as being inconsistent, unstructured, or entirely absent beyond the initial weeks post-partum. They recounted an initial focus on immediate post-natal recovery, but almost universally reported an absence of discussion on longer-term monitoring or follow-up care. They described communication gaps between maternity hospitals and primary care providers, particularly GPs, which may have led to uncertainty around post-pregnancy monitoring, and possible fragmentation of follow-up care. Some participants described how their GPs had not been aware of their preeclampsia diagnosis until directly informed by the patients themselves when they were attending post-natal appointments several weeks later, and this may have contributed to unclear follow-up pathways.

*"There was no communication between the hospital and the GP, which made it quite challenging... the connection between the obstetrician and the GP needs to be better"* [P9]

A small number of participants demonstrated pre-existing knowledge of links with heart disease or kidney disease, and they had typically discovered this information independently by *"Googling stuff"* [P11] shortly before the research interviews. However, most had been unaware of any associations with future risk of chronic disease. They were surprised by this information as they had not regarded preeclampsia as being relevant to their long-term health.

*"I was never really aware that there was any long-term effects. I thought it was just something that was associated with your pregnancy, and went away thereafter. But I don't really know anything else about it"* [P6]

Once participants were informed in the interview that preeclampsia may be relevant to their future risk of chronic disease, the majority of them perceived that this information was valuable to them. They described feeling disempowered by their information gaps, and that having this knowledge might *"plant a seed"* [P6] for their own personal decision-making around chronic disease prevention. Some participants also described the context of a broader need for women's health issues to be *"taken from the shadows"* [P4] in society, and for affected women to be trusted with honest and upfront information about their health.

*"I want total transparency. Give me the good, the bad and the ugly… you're putting the power in (my) hands to be able to do something about it" [P4]*

Only one participant described a preference to know less about these associations, stating that *"ignorance is bliss" [P3]* and that this information might give women undue cause for concern about their health. By contrast, most participants articulated the importance, and relevance, of receiving this information.

### 3.2 Changing priorities as *'life takes over'*

Participants described clear preferences around the optimal timing and format of information-sharing with them about long-term health risks. Most suggested that it would not be appropriate to inform them of future risks during the initial post-partum period, when their immediate priority was to recover from childbirth and take care of their newborn baby. Some participants described being *"bombarded with information" [P10]* immediately post-partum, and preference to discuss potential long-term risks a few months later when they had more *"headspace" [P8]*.

Participants suggested using routine postnatal appointments with the GP or public health/community nurse as initial signposting opportunities to flag that preeclampsia may be relevant as a risk marker for long-term health conditions. Beyond that, they expressed consistent preferences for one-to-one consultations focused on the long-term risk of chronic disease, ideally between six and 12 months postpartum.

*"I would love to think that maybe even a few months down the line, when you're over the chaos of a new baby, that there is an opportunity to have a discussion, and just to learn a little bit more about it" [P2]*

Participants expressed a desire for any discussions to be supplemented by physical, take-home reading material. They wanted this information to come from a trusted and reputable source, preferably endorsed by the health service, to *"stop (them) Googling" [P2]*.

Women broadly welcomed the concept of structured follow-up care. However, several of them described how time was a precious and limited resource, particularly during the initial years after pregnancy. Some tended to prioritise other family members' health needs before their own, which they felt might impact on women's uptake of personalised follow-up care, unless the rationale for attending was clearly communicated.

*"… nearly every woman will tell you that they can make an appointment for their husband. They'll make an appointment for their kids. They'll get the dog groomed, but they'll forget to go and get themselves checked out." [P1]*

Participants advocated for a model of follow-up care that is pragmatic in design and grounded in the everyday realities of mothers who often juggle multiple family and work-related responsibilities. They emphasised the need to consider accessibility and convenience of follow-up care, and expressed a clear preference for community-based follow-up care. They articulated the need for flexible appointment times, but also acknowledged that they would prefer in-person consultations instead of virtual follow-up.

*"…there's so much more involved than just simply getting into your car and going to the appointment. There might be lots of other factors to consider. Yeah, childcare and logistics and everything else." [P4]*

Notably, three participants had been diagnosed with preeclampsia during 2023, and were already eligible for follow-up through the expanded National Chronic Disease Prevention Programme in Ireland [12]. However, when probed about this, only one of them reported being enrolled in the programme to her knowledge, and she did not understand the purpose of this follow-up, as it had not been clearly explained using terminology which was accessible to her.

Several women expressed the need for follow-up care to be free at the point of access, without differentiation by private health insurance status. They felt that any out-of-pocket expenses would potentially exclude vulnerable women or lead them to de-prioritise appointments in the context of other financial demands.

*"not everybody has health insurance. And unless it's free, that will prevent women from going to pay for anything" [P12]*

### 3.3 Desire for proactive, *'blameless'* follow-up care

Some participants recognised the relevance that follow-up care, or post-preeclampsia monitoring, might have had in their own lives. One participant, who had been diagnosed with cardiovascular disease in the years after pregnancy, requiring insertion of multiple cardiac stents, articulated how such a programme might have helped to prevent or delay her own disease onset.

*"I never had any follow-up care which for me, looking at my past history now, perhaps, would have been a marker for what I ended up having" [P3]*

Some participants described instigating limited, ad-hoc follow-up care themselves (e.g., blood pressure checks) after preeclampsia. However, they preferred the idea of being enrolled in follow-up programmes by default, with reminders issued by the healthcare system, similar to systematic national screening programmes. They emphasised that follow-up programmes should not rely entirely on patient initiative. Instead, they wanted to feel supported and prioritised by the healthcare system, with invitations to participate through personalised text message, email, or phone call. One participant suggested that it may help if her obstetric history could be routinely flagged on her medical record alongside her other cardio-metabolic risk factors and comorbidities (such as smoking or diabetes), to serve as a reminder to healthcare providers that this may remain clinically relevant in the years after pregnancy.

*"…say you are 15 years down the line, and you're at the GP for something, it should still say preeclampsia on the file, so that the GP will be thinking … we'll just check your blood pressure" [P8]*

The need for person-centred follow-up care was expressed in various different ways. Women expressed a strong desire for long-term risk information to be delivered in an empathetic, *"blameless"* manner *[P8].* Some described a sense of shame, guilt, or even stigma relating to their preeclampsia diagnosis, particularly if they had struggled with their weight before or during pregnancy. They wanted health service messaging to remain non-judgmental, and for this to be framed around opportunities for secondary prevention.

Participants almost universally opposed the idea of any follow-up care being provided in a maternity hospital. Some participants volunteered that they had experienced adverse outcomes during pregnancy and were still living with the residual fear or trauma associated with those experiences, including one participant who had lost her baby shortly after delivery.

There were no strong preferences over whether follow-up care should be doctor-led or nurse-led, but participants expressed a strong desire for continuity of care, rather than meeting *"a revolving door" [P1]* of healthcare professionals and needing to recount their obstetric experiences every time.

Importantly, some participants described an ongoing sense of anxiety around attending medical appointments, and particularly blood pressure monitoring, after their experiences with preeclampsia. It was apparent from non-verbal cues during interviews (including tone of voice and facial gestures) that that this was a point of discomfort for some participants. In collaborative member reflection meetings, this finding was strongly endorsed and echoed by others, and it led to some restructuring of themes. Participants emphasised the importance of taking a sensitive, person-centred approach to maximise engagement in follow-up care.

*"…as soon as I see the blood pressure cuff, my heart starts racing, and this is 15 years later, and it still happens, so there could be women who would actually just not want to do that, they would want to avoid it"* [P10]

## 4. Discussion

### 4.1 Main findings

In Ireland, women affected by preeclampsia describe a lack of structured long-term follow-up care after their discharge from maternity services. Most participants in our study were unaware of any links between preeclampsia and future risk of chronic disease, and some were surprised to learn about these links. Nonetheless, most participants perceived this information to be valuable and empowering for them, because of its potential to influence their personal decision-making relating to chronic disease prevention. Participants described the importance of providing long-term health information in a non-judgmental, person-centred manner. Most expressed a desire to be enrolled in a structured follow-up care programme focused on chronic disease prevention, commencing several months after delivery, when they felt able to start prioritising their long-term health again. They wanted the health service to invite them to participate by default, and they emphasised the need for this to be delivered in an empathetic, sensitive manner.

Some findings from our study are consistent with previous research. In Denmark, a qualitative study of women affected by preeclampsia demonstrated low awareness of their future risk of cardiovascular disease, with participants seeking more detailed information about this in the context of other modifiable risk factors. They felt this information may empower them to reduce their heart disease risk [21]. Similarly, Australian women affected by hypertensive disorders of pregnancy endorsed the importance of individual risk counselling so they could better understand their personal likelihood of developing chronic diseases [22]. They wanted to receive individualised information supplemented by evidence-based, take-home reading materials which they could access at their own pace, similar to the preferences expressed by women in our study.

In a Norwegian qualitative study, women described high levels of variation in their individual experiences of post-pregnancy monitoring during the first two years after preeclampsia. Affected women tended to de-prioritise their personal health relative to that of other family members, like some women in our study [23]. Our participants emphasised the importance of pragmatic, flexible follow-up pathways, and a strong desire for the healthcare system to take a more proactive role in initiating follow-up care to maximise engagement or uptake. Similar preferences have been expressed in other international settings, including the need to send default follow-up invitations to women affected by preeclampsia [22], while reducing dependence on women advocating for their own healthcare needs [24]. Our participants' suggestion to incorporate obstetric history into patient records may also provide a practical step forward which should be transferable to other contexts.

Participants outlined concerns about communication gaps between maternity hospitals and GPs. This has been proposed as a possible reason for lower awareness of chronic disease risks among patients with preeclampsia in other settings [25, 26]. This gap may also be relevant to the enrolment of patients in systematic screening or follow-up pathways. Only one of the three participants in our study who was eligible for the new National Chronic Disease Prevention Programme in Ireland appeared to be enrolled in it. Enrolment depends on GP awareness of their patient's diagnosis of preeclampsia, among other factors, and in Ireland, there is no shared electronic health record in common use across primary and secondary care [14]. The absence of this shared record may hinder effective communication between obstetric and primary care services [27], and it may limit monitoring of attendance at post-partum follow-up appointments, as well as implementation of default follow-up reminders for eligible women [28].

Preeclampsia does not always arise in isolation, and it may be accompanied by other pregnancy-related complications such as preterm delivery or gestational diabetes, which may alter an individual's risk trajectory towards cardio-metabolic disease [1, 29, 30]. Until electronic record sharing becomes more embedded across health system structures in Ireland, there will be an ongoing need to encourage maternity care services to communicate comprehensive follow-up information to affected women and their primary care providers after pregnancy.

Preeclampsia remains a highly stressful experience for affected women [10, 31], and while some individuals quickly recover from the physical effects, the long-term psychological impact should not be under-estimated [32]. In our study, some women described traumatic experiences during the perinatal period, and a number of them lived with a residual sense of anxiety relating to their experience of preeclampsia several years later, particularly when attending medical appointments or undergoing blood pressure checks. This was evident in their strong support for any follow-up care to be based outside of acute hospitals, and it underlines the importance of providing person-centred, empathetic follow-up care in a location where they feel comfortable. These findings align with a Dutch qualitative study, where affected women expressed the need for enhanced follow-up care specifically for the emotional effects of preeclampsia [33]. These women have been described as a "forgotten group", and their unmet follow-up needs are likely to include aspects of both chronic disease prevention, and psycho-social support [34].

## 4.2 Strengths and limitations

This study included a varied group of women whose experiences of preeclampsia differed by timeframe, geographical location, and antenatal health insurance status. We considered that our sample of 12 participants provided information power, considering the broad aims of the study, where exploratory, cross-case analysis of participants would be undertaken [17]. Our sample provided a rich overview of collective experiences of follow-up care in Ireland, and may enhance transferability of findings to other contexts.

The lead researcher coded all transcripts and this provided greater depth and consistency to the analysis. However, the sole interviewer was a male medical doctor, and this may have introduced particular gender or education-based dynamics to sensitive discussions relating to maternal health. The use of analyst triangulation, and member reflection helped to identify blind-spots in the interpretation of the findings, and strengthened the transparency and trustworthiness of the analysis process, thereby increasing confirmability of the findings. For example, initial coding had resulted in proposed themes focusing on surface-level details of structured follow-up care, patient information gaps, and desirable components of follow-up care. However, through reflexive discussion with other members of the research team, and collaborative member reflection meetings, our themes were refined to also capture latent concepts underpinning the data. These included how maternal healthcare may be de-prioritised at an individual and health system level, and how some women continue to experience residual anxiety relating to their preeclampsia experience, which in itself may influence preferences for more sensitive, person-centred follow-up care.

Individual experiences of follow-up care may be difficult to separate completely from antenatal and perinatal care experiences. We did not explore these experiences in depth, nor did we ask about the number of times women had been diagnosed with preeclampsia. We did not ask any detailed questions about education or socio-economic status, and participants' expectations of follow-up care may have been impacted by these factors. Moreover, as outlined, some participants were learning about associations between preeclampsia and chronic disease for the first time during research interviews, and it is possible that these individuals may have been surprised or distracted by this information. This may have impacted on their expressed preferences in the context of real-time interviews without having time to reflect at length on this new information.

Since our study was promoted using institutional social media accounts, and directly by members of our research team, it is possible that some participants may have had higher levels of health literacy than others within the wider population. Three of the study participants were healthcare professionals, and this may have also coloured their own views of optimal follow-up care after preeclampsia.

## 5. Conclusion

Women affected by preeclampsia in Ireland appear to have low levels of awareness of links between this obstetric complication and long-term risks of chronic disease. Many participants viewed preeclampsia as having minimal impact on their

future health after pregnancy, and most had no experience of structured follow-up care after preeclampsia. They valued the opportunity to receive more detailed information about their individual risk of chronic disease, and they endorsed the importance of structured follow-up care after preeclampsia. They wanted any future model of care to be empathetic, person-centred, underpinned by streamlined communication between maternity services and primary care providers, and with default enrolment for all women diagnosed with preeclampsia.

## Supporting information

**S1 File. Topic guide.**
(DOCX)

**S2 File. Selected quotes illustrating responses to questions from topic guide.**
(DOCX)

## Acknowledgments

The authors would sincerely like to thank all women who participated in this research, and who shared their valuable insights and perspectives. We are grateful to all members of the PPI panel who advised on recruitment and dissemination of the findings herein. We also acknowledge colleagues in the INFANT Research Centre, University College Cork, who provided wide-ranging operational support with this research, particularly Michelle Dorgan, Jackie O'Leary, Nimisha Maria Vinod, Sonia Lenehan, Jerry Deasy, and Eoghan McKernan.

## Author contributions

**Conceptualization:** Peter M. Barrett.

**Data curation:** Peter M. Barrett.

**Formal analysis:** Peter M. Barrett, Aisling Jennings, Heike Roth.

**Funding acquisition:** Peter M. Barrett.

**Investigation:** Peter M. Barrett.

**Methodology:** Peter M. Barrett, Aisling Jennings, Heike Roth, Molly Byrne, Amanda Henry.

**Project administration:** Peter M. Barrett.

**Validation:** Aisling Jennings, Heike Roth.

**Writing – original draft:** Peter M. Barrett.

**Writing – review & editing:** Peter M. Barrett, Aisling Jennings, Heike Roth, Molly Byrne, Emma Wallace, Amanda Henry, Fergus P. McCarthy, Karolina Kublickiene, Ali S. Khashan.

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
