## [Decision Letter · Decision Letter 0]

19 Sep 2025

Dear Dr. Barrett,

Thank you for submitting your manuscript to PLOS ONE. After careful consideration, we feel that it has merit but does not fully meet PLOS ONE’s publication criteria as it currently stands. Therefore, we invite you to submit a revised version of the manuscript that addresses the points raised during the review process.

Thank you for submitting this interesting manuscript. The reviewers have raised several valid concerns and presented several suggestions. I would like to read your response to each of their points. Thank you.

We look forward to receiving your revised manuscript.

Kind regards,

Quetzal A. Class, PhD

Academic Editor

PLOS ONE

Journal Requirements:

Reviewers' comments:

Reviewer's Responses to Questions

**Comments to the Author**

1. Is the manuscript technically sound, and do the data support the conclusions?

Reviewer #1: Yes

Reviewer #2: Yes

2. Has the statistical analysis been performed appropriately and rigorously?

Reviewer #1: N/A

Reviewer #2: Yes

3. Have the authors made all data underlying the findings in their manuscript fully available?

Reviewer #1: Yes

Reviewer #2: Yes

4. Is the manuscript presented in an intelligible fashion and written in standard English?

Reviewer #1: Yes

Reviewer #2: Yes

Reviewer #1: This is a well conducted qualitative study on awareness of chronic disease risk and follow-up preferences after preeclampsia. The findings offer valuable insights into how postpartum follow-up can be tailored to meet women’s needs. I have only a few minor comments and suggestions for improvement.

Materials and methods:

- Page 4, line 20. Please clarify whether midwife-led public antenatal care in Ireland entirely lacks continuity of care, or if continuity is provided through consistent contact with the same midwife?

- Consider adding a brief description of the timing and structure of routine postpartum care in Ireland, either in the Introduction or the Methods section. This would help contextualize participants’ experiences and expectations.

- Page 5, line 12. Please describe how participant eligibility was assessed.

- Page 6 and 7. The description of the thematic analysis could benefit from additional detail. For example:

o What initial themes and subthemes emerged during the first round of coding?

o How did participant feedback influence the refinement or restructuring of these themes?

o Were any themes discarded or merged based on this feedback?

Results:

- Page 8, table 1. If available, please include information on participants’ parity and the severity of their preeclampsia as these factors may influence their postpartum experiences and perceptions of risk.

- Since some participants were recruited through the researchers’ own social media accounts, please clarify whether a significant proportion were healthcare professionals. If so, discuss how this may have influenced the results?

- Page 9, line 5. Please ensure consistent use of terminology in line with Braun and Clarke. Specifically, consider whether “themes” is more appropriate than “overarching themes” in this context, as the latter typically refers to broader conceptual categories not analyzed in depth.

- Page 9, lines 21-22. It would also be interesting to know whether study participants did not attend routine postpartum check-ups at all, or whether they attended check-ups but found that their history of preeclampsia was ignored.

Discussion:

- Page 15, line 10. The introductory statement regarding perinatal morbidity appears accurate but lacks a supporting reference. Additionally, as it is not a direct finding of this study, consider rephrasing or repositioning it.

- Please reflect on the general attendance rates for routine postpartum check-ups in Ireland and whether these differs between women with and without preeclampsia.

Reviewer #2: This manuscript by Barrett et al. set out to assess the post natal care experiences of women diagnosed with preeclampsia in Ireland. This is a qualitative study that recruited 12 patients using purposive and snowball sampling. The authors used semistructured interviews with embedded prompted questions to elicit patient knowledge of the postnatal complications of preeclampsia, patient experience with postnatal follow-up and what changes patients would have found beneficial in their postnatal follow-up. The authors successfully outlined the experiences of the women being interviewed and point to the key pitfalls that they experience: few of them were informed about the long-term health consequences of preeclampsia and many of them did not have structured postnatal follow-up. The authors also outlined the patients’ desire for streamlined enrollment to postnatal visits.

I agree with the premise that preeclampsia should not be “in the rear-view mirror” and the authors cite strong papers in support of that, but I think their argument on the importance of postnatal care in preeclamptic women can be strengthened with more details. Readers may want more detail regarding the strength of the relationship between preeclampsia and the longitudinal health concerns listed and whether there is data to support an impact on patient outcomes from postnatal follow-up.

The methodology of the study is well-justified, but a bit more detail on a few key points could enhance the reproducibility of the study. It would be useful to know more about the patient and public involvement panel, primarily, how they were selected. It is stated earlier in the manuscript that a diverse group of subjects from various insurance statuses, ages and locations were sought out; it is assumed that the panel was composed of a similarly diverse group, but this is not explicitly stated. Additionally, while it is stated that the subjects are from a diverse group of counties mixing rural and urban areas, this is not summarized in table 1. Inclusion of the mix of counties would further show the success of the researchers in establishing a diverse group of subjects.

Some additional details would be useful for understanding the results. Namely, the breakdown of participant answers for the prompted questions outlined in the appendices would be a useful addition to the well-summarized results. Such a table need not necessarily include the direct quotes of participants, but could be a breakdown of how responses were coded.

Some additional details would be useful for understanding the results. The results are outlined well and much detail is given on the subject’s responses, but a table might be useful so that readers can quickly see the results at a glance. Given that the interviewer had a list of prompted questions, the coded responses could make for an appealing table that summarizes the breakdown of response by question.

A few minor notes

In the methods section there is mention of the interviewer taking note of non-verbal cues of participants and that this impacted the final results. If possible, clarification of how results were impacted by these notes would be useful

In methods, it is discussed how preliminary results were shared with 3 of the subjects and that the subjects’ responses impacted the final analysis. Clarification of how results were impacted would be useful.

In the abstract there is mention of the diversity of postnatal care that patients experienced. The impression from reading the results section is that the subjects universally received inadequate postnatal care. Explicitly stating the type of diversity of care that patients experienced would support this point.

An overall limitation of the study that may want to be pointed out is the gender dynamics of the interviews. The sole interviewer in the study is male which might modulate the responses, given that this is a sensitive topic to do with women’s health.

**Do you want your identity to be public for this peer review?** For information about this choice, including consent withdrawal, please see our Privacy Policy

Reviewer #1: No

Reviewer #2: **Yes: ** Toma Tchernodrinski

---

## [Author Response · Author response to Decision Letter 1]

6 Nov 2025

Reviewer #1:

1. This is a well conducted qualitative study on awareness of chronic disease risk and follow-up preferences after preeclampsia. The findings offer valuable insights into how postpartum follow-up can be tailored to meet women’s needs. I have only a few minor comments and suggestions for improvement.

Many thanks for your positive feedback on our work.

2. Page 4, line 20. Please clarify whether midwife-led public antenatal care in Ireland entirely lacks continuity of care, or if continuity is provided through consistent contact with the same midwife?

Thank you for this suggestion. We have now added in some detail to clarify that this continuity of care is not routinely available for all women.

3. Consider adding a brief description of the timing and structure of routine postpartum care in Ireland, either in the Introduction or the Methods section. This would help contextualize participants’ experiences and expectations.

Thank you for this suggestion. Postpartum care in Ireland is shared between public health nurses, GPs, and obstetricians or midwives, and may vary between individuals. Public health nurses typically visit mothers at home during the first week after discharge from hospital, irrespective of their health insurance status. If they have any concerns about the mother or the baby, they will refer them to their GP or to the maternity hospital for further review/assessment. Mothers and their newborns are also encouraged to attend their GP at 2 weeks, and at 6 weeks postpartum for a holistic medical check-up. However, this appointment must be initiated by the mother and is not always offered by default. Those who have given birth through a private obstetric provider will typically have a check-up with their obstetrician at 4-6 weeks postpartum.

Beyond that, mothers who opt to avail of the routine childhood immunisation programme for their baby will attend either the GP or a practice nurse in the community at 2 months, 4 months, and 6 months postpartum, but the focus of those appointments is on the baby’s vaccines (although they do serve as another “touchpoint” with the health service). We have now added summary detail on this complex set of postpartum arrangements, including the potential for variation in individual patient experiences, to the Methods section.

4. Page 5, line 12. Please describe how participant eligibility was assessed.

Participant eligibility was assessed against our pre-defined inclusion criteria. Women needed to be diagnosed with preeclampsia at least once, more than one year prior to the interview, and they had to be currently residing in the Republic of Ireland, in order to be eligible to participate. We have now added this detail to the Methods.

5. Page 6 and 7. The description of the thematic analysis could benefit from additional detail. For example:

o What initial themes and subthemes emerged during the first round of coding?

o How did participant feedback influence the refinement or restructuring of these themes?

o Were any themes discarded or merged based on this feedback?

Thank you for this useful suggestion. This is a key area of the analysis, and we hope that the further detail provided here, and now within the manuscript, will be sufficient.

Initially we had generated three themes from the data, relating to (i) Lack of structured follow-up care (ii) Information gaps leading to patient disempowerment (iii) Desire for pro-active follow-up care after preeclampsia. As we continued with our coding we recognised that these themes were overly simplistic, semantic themes which only captured surface-level information. They did not fully capture the nuances and depth within the data, nor the latent themes within. This became clearer through reflexive discussions between the lead researcher and two other co-researchers (AJ, HR) who coded two transcripts each.

Moreover, it became clearer during collaborative member reflection meetings, that interviewees strongly endorsed some of the surface-level detail on residual anxiety relating to the prior diagnosis of preeclampsia, and the importance of providing person-centred follow-up care after preeclampsia. We used this additional information and context to explore this in greater depth and to refine our thematic analysis. We have provided further detail in the Methods (section 2.4) and Discussion (Strengths & Limitations) on the process of how we refined our themes, and how participant feedback influenced this process.

6. Page 8, table 1. If available, please include information on participants’ parity and the severity of their preeclampsia as these factors may influence their postpartum experiences and perceptions of risk.

We have added in a line to the initial paragraph of the Results section outlining that we did not ask directly about preeclampsia severity, nor about parity. However, we are aware, from participant responses, that they included a mix of primiparous and multiparous women and we have included this detail in the revised manuscript. We have also clarified in section 3.3 (focusing on the nature of their follow-up care) that we did not ask directly about adverse experiences during or after pregnancy, but that some women volunteered information in relation to this during interviews.

7. Since some participants were recruited through the researchers’ own social media accounts, please clarify whether a significant proportion were healthcare professionals. If so, discuss how this may have influenced the results?

Thank you for this suggestion. We did not specifically ask about occupational background, but most participants mentioned during interviews whether or not they were working outside of the home and, if so, what their professional background was. We are aware of three participants who were allied healthcare professionals, and it is possible that our recruitment methods may have influenced this. We have now added this detail to the Results section and to the Limitations section.

8. Page 9, line 5. Please ensure consistent use of terminology in line with Braun and Clarke. Specifically, consider whether “themes” is more appropriate than “overarching themes” in this context, as the latter typically refers to broader conceptual categories not analyzed in depth.

Thank you, we have now amended this.

9. Page 9, lines 21-22. It would also be interesting to know whether study participants did not attend routine postpartum check-ups at all, or whether they attended check-ups but found that their history of preeclampsia was ignored.

We did not directly ask whether participants attended all of their routine postpartum check-ups, since the structure of postpartum care may vary considerably between individuals. However, most participants referred to these appointments in the context of having attended them. We hope that the additional summary detail on how postpartum check-ups are structured in Ireland will provide helpful contextual information for this.

Notably, some participants described how their GP had been unaware of their preeclampsia diagnosis until they had informed the GP directly about it several weeks postpartum. This was typically due to suboptimal, or often delayed, communication between the maternity hospital and the GP. Even when this information had been communicated back to the GP, it did not appear to be routinely acted upon for the purposes of follow-up care, possibly due to lack of agreed, structured clinical care pathways. We have amended the wording in section 3.1 to try to make this point clearer.

10. Page 15, line 10. The introductory statement regarding perinatal morbidity appears accurate but lacks a supporting reference. Additionally, as it is not a direct finding of this study, consider rephrasing or repositioning it.

We agree with this point, and have removed this.

11. Please reflect on the general attendance rates for routine postpartum check-ups in Ireland and whether these differs between women with and without preeclampsia.

Thank you for this suggestion. As mentioned above, we have added some further detail and context on attendance at postpartum check-ups which may address some of this important point. Unfortunately, given the very fragmented information systems in primary care in Ireland, and the lack of a universal electronic health record, we are not aware of reliable data on attendance rates at routine postpartum check ups. Moreover, the structure and content of postpartum follow-up care may vary by health insurance status, among other factors. We have now added some more detail to section 4.1 on this limitation too.

Reviewer #2:

1. I agree with the premise that preeclampsia should not be “in the rear-view mirror” and the authors cite strong papers in support of that, but I think their argument on the importance of postnatal care in preeclamptic women can be strengthened with more details. Readers may want more detail regarding the strength of the relationship between preeclampsia and the longitudinal health concerns listed and whether there is data to support an impact on patient outcomes from postnatal follow-up.

Thank you for this helpful suggestion. We have endeavoured to add further detail on the strength of these associations in the Introduction, without providing too much technical, epidemiological detail. We have added further supporting references on previous studies conducted on subclinical atherosclerosis after preeclampsia, which is strongly suggestive of the need for early follow-up after pregnancy, as well as vascular dementia which provides a certain level of insight in to the long-term relevance of follow-up care.

The query whether there is data to support a clear impact of postnatal follow-up care on patient outcomes is an important but complex one. International clinical guidelines have almost universally emphasised the importance and relevance of providing follow-up care to reduce the long-term burden of cardiovascular disease, renal disease, and stroke. Yet, there is no clear consensus on how this should be optimally structured or implemented in practice, and to an extent, this may depend on how different health services are configured.

From a population health perspective, the relative risk of preeclampsia on these disease outcomes is quite clear (and we have added some summary detail to Section 1 in this regard), but the risk to individual patients may remain low/modest. To our knowledge, it is not yet clear what the optimal model of follow-up care should look like, or what might impact most on long-term patient outcomes. That is part of our reason for first exploring, in the Irish context, what might be feasible or acceptable to women affected by preeclampsia.

We have also added some detail in the Discussion to acknowledge that preeclampsia does not always arise in isolation, and is often accompanied by other pregnancy complications which may alter the individual risk towards cardio-metabolic disease. Ideally, these complications should be considered alongside preeclampsia when considering the follow-up care of impacted women.

2. The methodology of the study is well-justified, but a bit more detail on a few key points could enhance the reproducibility of the study. It would be useful to know more about the patient and public involvement panel, primarily, how they were selected. It is stated earlier in the manuscript that a diverse group of subjects from various insurance statuses, ages and locations were sought out; it is assumed that the panel was composed of a similarly diverse group, but this is not explicitly stated.

Thank you for this suggestion. We have now added detail on the PPI panel to the Methods section. A new PPI panel was convened to support this study, and as part of a larger research project on the long-term impacts of preeclampsia on maternal chronic disease risk. This panel was established in 2024 by the lead researcher. A public-facing advertisement was posted on the PPI Ignite website, and promoted through social media posts, inviting women affected by preeclampsia (at least one year prior) to participate in an advisory group. (* Preeclampsia during pregnancy and the long-term prevention of chronic disease in affected women • PPI Ignite Network)

The PPI panel comprises five members who represent a mix of ages, geographical locations in Ireland, parity, and public/private health insurance status. One of the PPI contributors has a physical disability and offers a different perspective from other contributors too (although we would prefer not to include that detail in the manuscript out of respect to her).

3. Additionally, while it is stated that the subjects are from a diverse group of counties mixing rural and urban areas, this is not summarized in table 1. Inclusion of the mix of counties would further show the success of the researchers in establishing a diverse group of subjects.

Thank you for this suggestion. We agree that this information would more clearly illustrate our success in establishing a diverse group of participants. Although we did collect information on county of residence, we did not present this in the manuscript because of the possibility of some individuals becoming identifiable in the context of their age, time since diagnosis, insurance status, and comorbidities (particularly in the Irish context where some counties have a reasonably small population). We have now added a line to the results section to clarify that seven of the participants lived in urban or sub-urban areas, while five participants lived in more rural areas. We have also added this information in summary format to Table 1 and we hope that this will be considered sufficient. Further information on county of residence may be provided to the reviewer upon request, but we do not believe that it would be appropriate to publish this, unless we omit some other information on age, time since diagnosis, or comorbidity.

4. Some additional details would be useful for understanding the results. Namely, the breakdown of participant answers for the prompted questions outlined in the appendices would be a useful addition to the well-summarized results. Such a table need not necessarily include the direct quotes of participants, but could be a breakdown of how responses were coded.

Thank you for this suggestion. The topic guide was used in the context of semi-structured interviews, where the exact questions posed to participants varied in accordance with the nature of the discussion. Some participants were asked all questions within the topic guide, whereas others were not, depending on the nature of the discussion. We have now added a summary of sample responses to selected question areas covered within the topic guides, and this has been provided as a supplementary file in Appendix 2. It would not be possible to provide a comprehensive overview of all responses, since the interviews generated 139 pages of transcripts, and a very extensive number of codes. However, we hope that the summary detail provided will help with the overall understanding and context of the Results.

5. Some additional details would be useful for understanding the results. The results are outlined well and much detail is given on the subject’s responses, but a table might be useful so that readers can quickly see the results at a glance. Given that the interviewer had a list of prompted questions, the coded responses could make for an appealing table that summarizes the breakdown of response by question.

Thank you for this suggestion and for the positive feedback on our Results section. As mentioned above, while the interviewer had a list of approximately 18 prompted questions (depending on the nature and direction of the discussion), the exact questions posed to each individual participant differed as the format of the interview was semi-structured. If a participant had volunteered some relevant information in an unprompted manner, then they were not specifically asked each of the prompt questions. Equally, if a participant communicated any discomfort or sensitivity around a particular topic (for example, due to an adverse health outcome) they were not asked questions which might have

---

## [Decision Letter · Decision Letter 1]

17 Nov 2025

“You just forget about preeclampsia and move on” –awareness of chronic disease risks and follow-up preferences after preeclampsia in Ireland: a national qualitative study.

PONE-D-25-44554R1

Dear Dr. Barrett,

We’re pleased to inform you that your manuscript has been judged scientifically suitable for publication and will be formally accepted for publication once it meets all outstanding technical requirements.

Kind regards,

Quetzal A. Class, PhD

Academic Editor

PLOS ONE

Additional Editor Comments (optional):

Reviewers' comments:

Reviewer's Responses to Questions

**Comments to the Author**

Reviewer #1: All comments have been addressed

Reviewer #2: All comments have been addressed

2. Is the manuscript technically sound, and do the data support the conclusions?

Reviewer #1: Yes

Reviewer #2: Yes

3. Has the statistical analysis been performed appropriately and rigorously?

Reviewer #1: N/A

Reviewer #2: Yes

4. Have the authors made all data underlying the findings in their manuscript fully available?

Reviewer #1: (No Response)

Reviewer #2: Yes

5. Is the manuscript presented in an intelligible fashion and written in standard English?

Reviewer #1: No

Reviewer #2: Yes

Reviewer #1: All of my comments have been satisfactorily addressed, and the manuscript has been revised accordingly.

Reviewer #2: I am happy with the changes made to the manuscript based on my suggestions. They went above and beyond in responding to my points of concern. I have no other concerns and am happy with the shape of this manuscript.

**Do you want your identity to be public for this peer review?** For information about this choice, including consent withdrawal, please see our Privacy Policy

Reviewer #1: No

Reviewer #2: **Yes: ** Toma Tchernodrinski

---

## [Editor Report · Acceptance letter]

PONE-D-25-44554R1

PLOS One

Dear Dr. Barrett,

I'm pleased to inform you that your manuscript has been deemed suitable for publication in PLOS One. Congratulations! Your manuscript is now being handed over to our production team.

Kind regards,

on behalf of

Dr. Quetzal A. Class

Academic Editor

PLOS One